# Emerging practices supporting diabetes self-management among food insecure adults and families: A scoping review

Enza Gucciardi[1]*, Adalia Yang[1‡], Katharine Cohen-Olivenstein[1‡], Brittany Parmentier[1‡], Jessica Wegener[1☯], Vanita Pais[2☯]

**1** School of Nutrition, Ryerson University, Toronto, Ontario, Canada, **2** Division of Endocrinology, Hospital for Sick Children, Toronto, Ontario, Canada

☯ These authors contributed equally to this work.
‡ These authors also contributed equally to this work.
* egucciar@ryerson.ca

**Data Availability Statement:** All relevant data are within the manuscript and its Supporting Information files.

## Abstract

### Background

Food insecurity undermines a patient's ability to follow diabetes self-management recommendations. Care providers need strategies to direct their support of diabetes management among food insecure patients and families.

### Objective

To identify what emerging practices health care providers can relay to patients or operationalize to best support diabetes self-management among food insecure adults and families.

### Eligibility criteria

Food insecure populations with diabetes (type 1, type 2, prediabetes, gestational diabetes) and provided diabetes management practices specifically for food insecure populations. Only studies in English were considered. In total, 21 articles were reviewed.

### Sources of evidence

Seven databases: Cumulative Index of Nursing and Allied Health Literature, Cochrane Database of Systematic Reviews, Medline, ProQuest Nursing & Allied Health Database, PsychInfo, Scopus, and Web of Science.

### Results

Emerging practices identified through this review include screening for food insecurity as a first step, followed by tailoring nutrition counseling, preventing hypoglycemia through managing medications, referring patients to professional and community resources, building supportive care provider-patient relationships, developing constructive coping strategies, and decreasing tobacco smoking.

**Funding:** A grant from the Lawson Foundation (GRT 2016-110) was awarded to EG. https://lawson.ca/. The funders had no role in study design, data collection and analysis, decision to publish, or preparation of the manuscript.

**Competing interests:** The authors have declared that no competing interests exist.

## Conclusion

Emerging practices identified in our review include screening for food insecurity, nutrition counselling, tailoring management plans through medication adjustments, referring to local resources, improving care provider–patient relationship, promoting healthy coping strategies, and decreasing tobacco use. These strategies can help care providers better support food insecure populations with diabetes. However, some strategies require further evaluation to enhance understanding of their benefits, particularly in food insecure individuals with gestational and prediabetes, as no studies were identified in these populations. A major limitation of this review is the lack of global representation considering no studies outside of North America satisfied our inclusion criteria, due in part to the English language restriction.

## Introduction

Food insecurity persists among North Americans with diabetes [1–5]. Food insecurity refers to inadequate or insecure access to food due to financial constraints [1]. In 2005, the prevalence of food insecurity in Canada was 9.3% among households with individuals with diabetes, compared to 6.8% among households without [2]. The likelihood of food insecurity increases by 4% with every year earlier an individual is diagnosed with diabetes [2]. For instance, in Nova Scotia, food insecurity prevalence is substantially higher in households with a child with diabetes (21.9%) than with households with only an adult with diabetes (14.6%), suggesting higher risks associated with food insecurity among households with children with diabetes [6]. Persons with pre-diabetes are 39% more likely to experience food insecurity [7,8] as food insecurity of any degree has been shown to increase the risk of pre-diabetes. The likelihood of gestational diabetes is also higher in women who are considered marginally food insecure [9] as pregnant women who are food insecure experience greater weight gain during pregnancy and are more likely to be obese prior to becoming pregnant [9].

A few studies outside of North America have identified a higher prevalence of food insecurity among those with diabetes or have identified food insecurity as a risk factor for poorer diabetes management. For instance, a study in Iran showed that those who were food insecure were 2.8 times more likely to have diabetes than those who were food secure [10]. In Kenya, food insecure individuals with diabetes were more likely to be on insulin or have had been on insulin compared to their food secure counterparts [11]. Another study reported severely food insecure Jordanians with diabetes had a higher body mass index (BMI) despite having a lower caloric intake than food secure or mildly food insecure individuals with diabetes [12].

Healthy eating is key to diabetes prevention and management in both adults and children. However, food insecure individuals with diabetes often eat fewer fruits and vegetables [2] and have poorer quality diets that are low in variety [13,14]. Food insecurity undermines individuals' ability to purchase and consume recommended foods and follow self-management plans. Growing literature links food insecurity with poor diabetes self-management, adverse health outcomes, and increased healthcare costs. Food insecurity is associated with poor glycemic-monitoring adherence [15], increased likelihood of poor glycemic control [16,17], and higher rates of hospitalization and use of health services [6,15,18]. In children, poor glycemic control can cause hypoglycemia and ketoacidosis, leading to hospital admissions and long-term consequences: retinopathy, nephropathy, neuropathy, and increased risk of cardiovascular disease [6]. Food insecure adults with diabetes report more cost-related medication underuse and

poor adherence to oral hypoglycemic agents [19,20]. Additionally, they report skipping meals, eating more energy-dense foods and foods higher in sodium, and have higher levels of diabetes-related emotional distress [21]. Food insecure individuals are more likely to describe their mental health, satisfaction with life, and self-perceived stress in neutral or negative terms [2,22]. Given the link between food insecurity and poor health outcomes for individuals with diabetes, it is not surprising that annualized total American healthcare expenditures on food insecure individuals with diabetes are estimated to be US $4,414 higher than their food secure counterparts [23].

Households that are food insecure may be ill-equipped to successfully manage diabetes, as financial strain and competing priorities often force them to cut expenses on diabetes medication and supplies and healthy foods to meet housing costs [21]. Literature now recommends routine screening for food insecurity among individuals with diabetes [5]. This screening can help clinicians tailor diabetes-management plans for food insecure individuals and may significantly reduce medical costs [24–26]. For instance, food insecurity knowledge helps clinicians provide patients with more realistic dietary recommendations [5]; identify patient difficulties in adhering to prescribed medications [27]; and identify patients at increased risk of poor health outcomes associated with food insecurity (e.g., asthma, depression, obesity) [28]. However, for routine screening to succeed, care providers must have guidelines on how best to support diabetes management among food insecure patients and families. No such guidelines exist.

The primary aim of this scoping review is to identify recommendations or emerging practices that health care providers can relay to patients or operationalize to support diabetes self-management among food insecure populations.

To our knowledge, this is the first scoping review to investigate emerging practices to support diabetes self-management in the context of food insecurity in both adult and pediatric populations. Identified emerging practices are not intended as solutions to food insecurity. Instead, our aim is to better support diabetes management among food insecure populations with diabetes.

## Methods

This scoping review seeks to answer the question: What recommendations or emerging practices are being conveyed to patients or used by healthcare providers to support diabetes self-management among food insecure populations?

This paper will define emerging practices as recommendations, practices, strategies or "interventions that are new, innovative and which hold promise based on some level of evidence of effectiveness or change that is not research-based and/or sufficient to be deemed a 'promising' or 'best' practice" yet [29]. As such, practices that are currently in use but have yet to be substantially evaluated have been included. Emerging practices must also be based on "protocols, standards, or preferred practice patterns that [may] lead to effective–health outcomes" [30].

### Eligibility

For all searches, studies were included or excluded based on the Population, Concept, and Context (PCC) framework for scoping reviews [31]. As such, the participant population was defined as food insecure populations with diabetes (prediabetes, type 1 or 2, or gestational); the concept was recommendations, practices, strategies, or interventions of any nature that addressed diabetes self-management in a food insecure population; studies of all contexts were

considered with no specifications for timing and setting. Studies of all designs were acceptable. The studies needed to be published in English for review.

## Data sources and search strategy

We conducted a scoping review focusing on diabetes populations who are food insecure following the guidelines recommended in the PRISMA extension for scoping reviews checklist [32]. Seven databases were electronically searched: Cumulative Index of Nursing and Allied Health Literature (CINAHL), Cochrane Database of Systematic Reviews, Medline, ProQuest Nursing & Allied Health Database, PsychInfo, Scopus, and Web of Science. Combinations of the following key words were used: diabetes, diabetes mellitus, type 1 diabetes, diabetes mellitus, type 1, type 2 diabetes, diabetes mellitus, type 2, gestational diabetes, gestational, prediabetes, prediabetic state, food security, food insecurity, food supply, cooking, food skills, education, patient education, health education, coping strategies, therapeutics, self-efficacy, diabetes management, self-management, self-care, low income, poverty, hunger, pediatric, newborn, infant, preschool child, child, adolescent, family characteristic, family, and household. See Table 1 for search strategy used. Additional articles were found through bibliography hand searching and expert consultation.

## Study selection

The search conducted for all dates up to November 2018, retrieved 3066 articles (Fig 1). Seven additional articles were found through bibliography hand searches and expert consultation. Two reviewers independently screened through article titles and abstracts using DistillerSR. Acceptable articles were reviewed in full to confirm eligibility and extract relevant information for the scoping review. Twenty-one articles satisfied inclusion criteria and were reviewed. Any discrepancies were resolved through discussion. Studies were most often excluded because

**Table 1. Search strategy for Ovid MEDLINE.**

| # | Search | Results |
|---|--------|---------|
| 1 | (diabetes mellitus OR diabetes mellitus, type 1 OR diabetes mellitus, type 2 OR diabetes, gestational OR prediabetic state) AND (food supply OR food security) | 171 |
| 2 | (diabetes mellitus OR diabetes mellitus, type 1 OR diabetes mellitus, type 2 OR diabetes, gestational OR prediabetic state) AND (food supply OR food security) AND education | 7 |
| 3 | (diabetes mellitus OR diabetes mellitus, type 1 OR diabetes mellitus, type 2 OR diabetes, gestational OR prediabetic state) AND (food supply OR food security) AND skills | 1 |
| 4 | (diabetes mellitus OR diabetes mellitus, type 1 OR diabetes mellitus, type 2 OR diabetes, gestational OR prediabetic state) AND (food supply OR food security) AND cooking | 0 |
| 5 | (diabetes mellitus OR diabetes mellitus, type 1 OR diabetes mellitus, type 2 OR diabetes, gestational OR prediabetic state) AND (cooking OR food skills) | 286 |
| 6 | (diabetes mellitus OR diabetes mellitus, type 1 OR diabetes mellitus, type 2 OR diabetes, gestational OR prediabetic state) AND (Child OR Child, Preschool OR Infant OR Infant, Newborn, OR Adolescent) | 31 |
| 7 | (diabetes mellitus OR diabetes mellitus, type 1 OR diabetes mellitus, type 2 OR diabetes, gestational OR prediabetic state) AND poverty AND Patient education as topic | 5 |
| 8 | (diabetes mellitus OR diabetes mellitus, type 1 OR diabetes mellitus, type 2 OR diabetes, gestational OR prediabetic state) AND (food security OR food supply) AND family | 7 |
| 9 | (diabetes mellitus OR diabetes mellitus, type 1 OR diabetes mellitus, type 2 OR diabetes, gestational OR prediabetic state) AND (food security OR food supply) AND (family characteristics OR household) | 31 |
| 10 | (diabetes mellitus OR diabetes mellitus, type 1 OR diabetes mellitus, type 2 OR diabetes, gestational OR prediabetic state) AND (food security OR food supply) AND therapeutics | 20 |

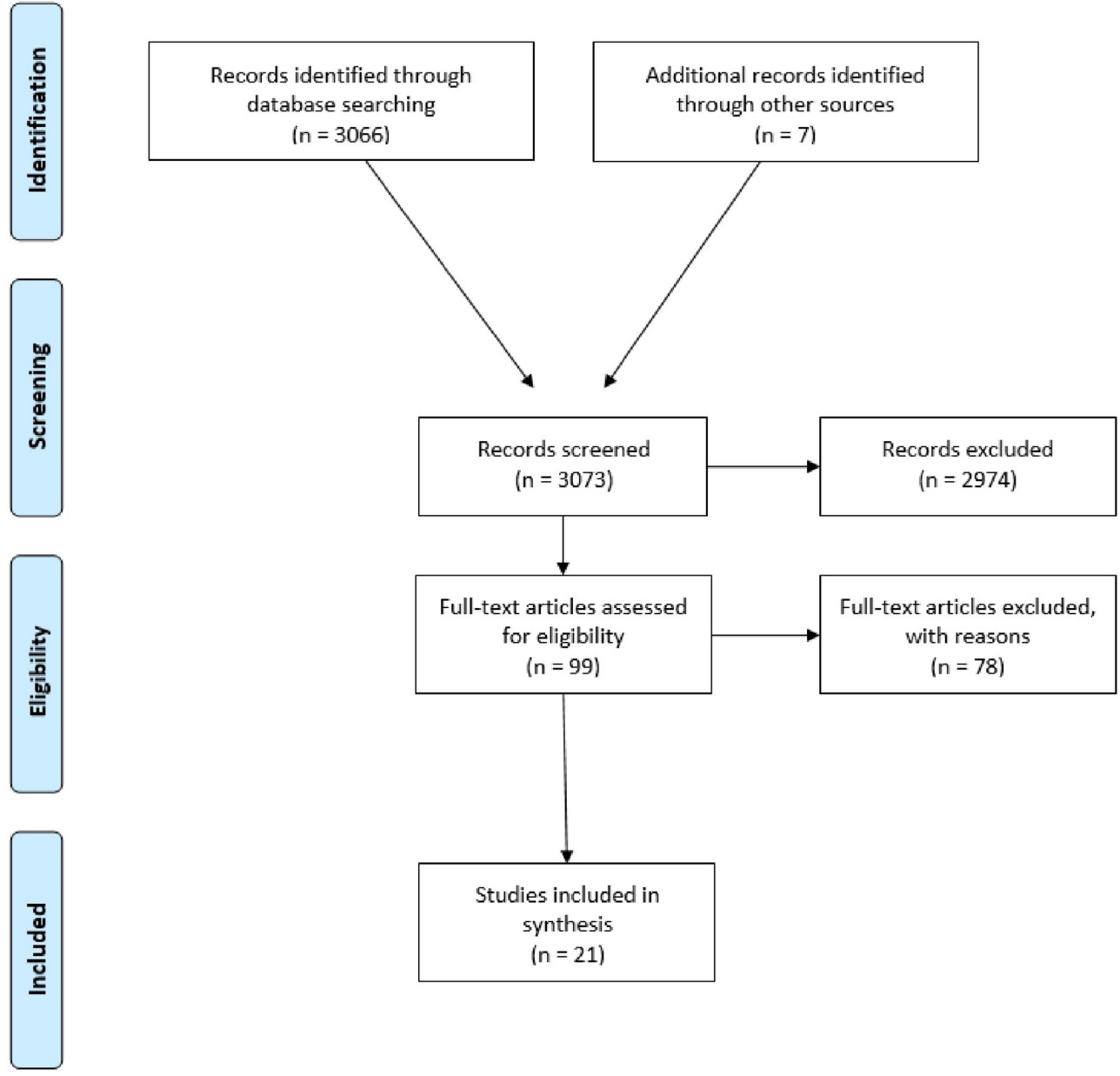

**Fig 1. PRISMA chart outlining data study selection process.**

they were not specific to a food insecure population with diabetes, or did not discuss diabetes-management practices, strategies, or interventions for those who are food insecure that can be operationalized by care providers. Only full text-articles were included in this review.

## Data analysis and synthesis

For each article reviewed in full, the reference and publication information, objectives, study design and methods, target population and sample size, main results, and emerging practices, strategies, on interventions were extracted, see Tables 2 and 3 for study characteristics. The emerging practices to support diabetes self-management were compiled and first organized

**Table 2. Characteristics of included studies conducted in pediatric population.**

| Reference | Objectives | Study Design & Methods | Target Population & Sample Size | Results | Recommendations for Care Providers |
|---|---|---|---|---|---|
| Protudjer et al., 2014 [35] (Canada) | Describe lived experiences of youth with type 2 diabetes from point of view of youth, caregivers, and healthcare professionals. Identify barriers and facilitators to lifestyle approaches to diabetes self-management in a low-income context. Generate a grounded theory. | Qualitative- Grounded theory approach Use of an inductive approach, open-ended questions, purposive sampling. | Interviews and focus groups with 8 youth with type 2 diabetes aged <18 years of age, 6 primary caregivers, and 8 healthcare professionals from a pediatric endocrinology clinic in Canada. | Supportive relationships are important determinants of lifestyle approaches to diabetes self-management, according to youth and primary caregivers. All 3 groups identify social determinants of health (food insecurity, poverty) as major barriers. Barriers for type 1 and type 2 diabetes differ according to healthcare professionals. | a) More regular contact with healthcare professionals b) Cultural competency training for healthcare professionals c) Facilitate access to recreational facilities to support physical activity |
| Marjerrison et al., 2011 [6] (Canada) | Examine the prevalence of food insecurity in households with a child with insulin-requiring diabetes mellitus (DM), compared to provincial and national prevalence. Explore the association between food insecurity and suboptimal DM control, as measured by A1C and hospital admissions. Describe household characteristics and coping strategies of food insecure families with a child with DM. | Cross-sectional Data collected via telephone-administered questionnaire, 18-item Household Food Security Survey (HFSS) Module of the Canadian Community Health Survey (CCHS), clinical data from medical records. | 183 children < 18 years of age recruited from 2 general pediatric practices in Nova Scotia, Canada. | 21.9% of families with a child with diabetes were found to be food insecure, compared with the overall prevalence of 14.6% in Nova Scotia and 9.2% in Canada. Univariate analysis revealed food insecurity was associated with higher A1C (9.5% ± 2.13%, p<0.039). Multivariate analysis revealed child's age OR 1.115 (95% confidence interval [CI], 1.030–2.207) and parent's education OR 0.396 (95% confidence interval [CI], 0.167–0.819) were independent predictors of A1C. Common coping strategies include buying less expensive food, having another family member eat less, and reusing DM supplies. | a) Screen families with a child with DM for food insecurity b) Provide families with financial counseling/support c) Advocate for community resources to support children with DM and their families who are food insecure |
| Vitale et al., 2019 [34] (Canada) | To examine the acceptability and feasibility of a food insecurity screening initiative for families with a child with diabetes from the point of view of care providers and families. Also, to reveal facilitators and barriers to incorporating food insecurity screening into practice. | Pilot Study/Grey Literature The screening initiative was comprised of a food insecurity screening questionnaire, a care algorithm tailored to patient's needs, a handout outlining community resources, and a poster to increase awareness and reduce stigma around food insecurity. | 3 Canadian diabetes dietitian educators incorporated the screening initiative. 50 families were screened for food insecurity, and 37 of those families participated in an interview to discuss their experience. | Most families and care providers reported feeling comfortable with the screening initiative, however, having a provider-patient relationship increased care provider's willingness to screen patients. A major barrier to food insecurity screening was time constraints–care providers did not want to screen positively for food insecurity but not have sufficient time to discuss potential resources and options. | a) Food insecurity screening can provide clinicians with important information to tailor care and recommend appropriate resources to patients. b) Using motivational interviewing techniques, communicating nonjudgmentally, and asking patients to complete screening before appointment or using self-completed questionnaires may increase care provider comfort with food insecurity screening, particularly when trust and rapport between patient and clinician have not yet been established. c) Self-completed food insecurity questionnaires can be used under time constraints |

**Table 3.** Characteristics of included studies conducted in adult population.

| Reference | Objective | Study Design & Methods | Target Population & Sample Size | Results | Recommendations for Care Providers |
|---|---|---|---|---|---|
| Barnard et al., 2015 [4] (USA) | Summarize the current literature regarding interventions that provide material support for income, food, housing, and other basic needs | Review Searched National Library of Medicine's PubMed, PsycINFO, and CINAHL. Search terms included diabetes mellitus/ therapy, food supply, housing, medication adherence, poverty, socioeconomic factors, income, public assistance/statistics & numerical data, delivery of care/ economics. Included studies that described interventions or evaluations of programs that addressed income, food, and housing support on diabetes outcomes. | Adults aged 18 years and older with diabetes and food insecurity and/or housing instability. | Categorized interventions under food-, housing-, medication-, or income-based. Food: farmer's market vouchers, food prescriptions. Medication: better coverage in Canada for those <65. Interventions to support food, housing, and income may prevent diabetes and lower diabetes-related mortality. | a) Food and housing are important targets for clinical outcomes. |
| Berkowitz & Fabreau, 2015 [24] (USA) | | Commentary | Chronic disease management, particularly cardio-metabolic diseases, with a focus on diabetes. | Discussing food insecurity with patients is appropriate when it will change clinical management and may make care more patient-centered. Medications can be adjusted when food is limited. Culturally appropriate nutrition advice and community programs to connect patients with nutritional assistance. Examples include Community Action Programs in Toronto and Improving Diabetes Outcomes in Chicago, cooking classes, education and empowerment, food prescriptions. | a) Screening for food insecurity is appropriate when it may impact clinical management b) Connect patients to communication nutritional assistance programs. |
| Chan et al., 2015 [25] (Canada) | Explore how food insecurity affects individuals' ability to manage their diabetes | Qualitative Deductive thematic analysis of qualitative interviews | 21 English-speaking adults diagnosed with diabetes, who experienced food insecurity within the past year, as defined by 3 food insecurity screening questions. Patients were recruited from various community health centers in Toronto, Ontario that serve a low-income population. | Three themes emerged from analysis of participants' experiences of living with food insecurity and diabetes: (1) barriers to accessing and preparing food, (2) social isolation, and (3) enhancing agency and resilience. Food insecurity appears to negatively impact diabetes self-management. | a) Screen for food insecurity and refer patients to RD b) Clinicians should focus on reducing portion size of foods that are available and accessible to patients, rather than focusing on food and beverage substitutions that may not be attainable c) Keep list of meal delivery services d) Prescribe medications with low risk for hypoglycemia e) Use empathetic, individual approach |

(Continued)

**Table 3.** (Continued)

| Reference | Objective | Study Design & Methods | Target Population & Sample Size | Results | Recommendations for Care Providers |
|---|---|---|---|---|---|
| Essien et al., 2016 [36] (USA) | Summarize the current literature regarding the relationship between type 2 diabetes risk, diabetes control, and food insecurity. Explain underlying mechanisms. | Review Searched National Library of Medicine's PubMed using search terms diabetes mellitus, prediabetic state, abdominal obesity, food insecurity, food insufficiency, food supply, hunger, poverty. Included longitudinal, cross-sectional, and interventional designs. | Included individuals aged 12–75 with diabetes and/or abdominal obesity faced with food insecurity, hunger, or poverty. | Food insecurity is associated with diabetes risk factors such as poor diet quality, obesity (p<0.002), inflammation OR 1.21 (95% confidence interval [CI], 1.04–1.40), central adiposity (p<0.001), prediabetes, and insulin resistance. Food insecure individuals with diabetes were found to have poorer diabetes control and self-management skills, and increased diabetes complications. | a) Dietary habits and food relief programs are important areas for intervention to enable food access and to ease competing demands (medication, housing, etc.) to improve clinical outcomes. b) Educational diabetes self-management support interventions. |
| Galesloot et al., 2012 [5] (Canada) | To examine the prevalence of adult-level household food insecurity among clients receiving outpatient diabetes healthcare services. | Cross-sectional During client's scheduled session, clinicians administered the 10-item Household Food Security Survey Module (HFSSM) to determine severity of food insecurity. | Surveyed 314 adult patients with diabetes receiving care in a clinic over a 4-month period. | The prevalence of adult-level household food insecurity was 15.0% (95% confidence interval [CI], 11.2 to 19.4). Of clinic attendees, 6.7% (95% CI, 4.2 to 10.0) were categorized as severely food insecure. Comparable results from Alberta in 2007 using the same HFSSM instrument were 5.6% and 1.2%, respectively. | a) Formulation of realistic dietary plan with a focus on supporting access to food and diabetes supplies. b) Ask clients how they are currently dealing with food insecurity, reinforcing nutritionally sound substitutions and making interventions cost-effective. c) Print resources that acknowledge that healthy eating is difficult for people who have diabetes and live in food insecure households. |
| Gucciardi et al., 2014 [21] (Canada) | To synthesize the current literature on food insecurity and diabetes self-management. | Review Searched Medline, CINAHL, Cochrane Database of Systematic Reviews, Web of Science, and PyschINFO using search terms diabetes, diabetes mellitus, dietary intake, food access, food deserts, food insecurity, food security, food intake, food preferences, food supply, self-care, self-efficacy, self-management. Articles included if published before May 2014. | Reviewed 39 articles that explored or measured food insecurity, food augmentation strategies, food access, and/or dietary intake in a population with diabetes. | Summarizes effects of food insecurity and diabetes, as well as recommendations for healthcare providers, screening for food insecurity, nutritional counseling, medications for reducing hypoglycemia, diabetes self-management education. | a) Include food insecurity screening in diabetes patient assessment b) Screening for food insecurity should be non-judgmental to reduce risk of patients feeling embarrassed or ashamed. c) Consider patient's resources for purchasing, preparing, and cooking food in nutritional counseling. d) Explore challenges that may hinder patients' ability to follow therapeutic diets. e) Help patients identify resources within their communities. |
| Gundersen & Seligman, 2017 [37] (USA) | To summarize the extent of food insecurity, underlying determinants of food insecurity and potential health consequences, and several promising approaches to decrease food insecurity and related health issues. | Review | NA | Food insecurity is a highly prevalent health crisis that results in poor health outcomes (i.e. poor mental health, poor oral health, greater number of hospitalizations, etc.) and, resultantly, high health care expenditure. Two approaches that address food insecurity and its associated health consequences are the supplemental nutrition assistance program (SNAP) and diabetes-tailored food provision and diabetes education through food pantries and food bank partnerships. | a) Consider the use of food banks as a setting to provide diabetes education and nutritious foods appropriate for individuals with diabetes. b) Encourage food insecure patients to enroll in food assistance programs similar to SNAP to decrease food expenditure. |

(Continued)

 

Table 3. (Continued)

| Reference | Objective | Study Design & Methods | Target Population & Sample Size | Results | Recommendations for Care Providers |
|---|---|---|---|---|---|
| Ippolito et al., 2016 [41] (USA) | Examine the association between food insecurity and diabetes self-management in food pantry clients | Cross-sectional Descriptive Study Measures include 6-Item Household Food Security Survey (HFSS) Module, A1C, diabetes self-efficacy, diabetes distress, medication non-adherence, severe hypoglycemia, depressive symptoms, medication affordability, and food-medicine purchasing trade-offs. | Convenience sample of adults ≥ 18 years of age with diabetes at food pantries in California, Ohio, and Texas. | Significantly poorer diabetes self-management in food insecure groups, compared with food secure groups (p<0.001). Food insecure populations had 0.51 lower diabetes self-efficacy score (95% confidence interval [CI], -0.85 – -0.17), 0.79 greater diabetes distress score (95% confidence interval [CI], 0.54–1.04), medication non-adherence scores 0.31 higher (95% confidence interval [CI], 0.12–0.50), higher prevalence of severe hypoglycemia (OR 2.63 (95% confidence interval [CI], 1.42–4.85). Significantly higher prevalence of depressive symptoms, medication affordability challenges, and food medicine and health supply trade-offs. | a) Deliver healthcare and self-management support services and prescription food programs through food pantries, because food pantry users with diabetes may not seek clinical care as often as food secure counterparts. b) Train patients to manage diabetes medications in settings of reduced dietary intake. |
| Knight et al., 2016 [27] (USA) | Examine the prevalence of food insecurity in adults with diabetes. Determine the association between food insecurity and cutting back on prescribed medications due to financial constraints (i.e., scrimping). | Cross-sectional Data obtained from 2011 National Health Interview Survey. Food security status determined by the 10-item USDA food insecurity scale. | Study included 3242 adults from the United States with self-reported diabetes. | Approximately 17% of adults with diabetes in the NHIS survey were found to be food insecure. An additional 8.8% were found to be marginally food insecure. Of respondents with diabetes, 18.9% reported medication scrimping: 11.7% of food secure (FS) individuals, 27.7% of marginally food secure (MFS) individuals, and 45.6% of food insecure (FI) individuals. MFS and FI are strongly associated with scrimping (p<0.0001) in adjusted analyses. | a) Food security screening during appointments to identify patients who require assistance referrals and are unable to adhere to prescribed medication regimes. b) Provide referrals to supplemental food programs and/or community food resources to support food insecure individuals with diabetes. c) As part of dietary counseling, provide nutritional and financial counseling for individuals at higher risk of food insecurity. |
| Lopez & Seligman, 2012 [44] (USA) | | Commentary | Discusses screening of those with risk factors: ethnic minorities, low income, low education, single parents. | Screening questions: essential to ask questions in a non-judgmental way. Addressing hyperglycemia: keep list of local food resources (meals on wheels, food banks, soup kitchens), enroll children in school meal programs. Smoking cessation: increased smoking cessation support for family members could relieve pressure on food budgets. | a) Focus on decreasing portion sizes of financially/geographically available foods, rather than substituting foods. b) Eat out less, purchase frozen fruits and vegetables, buy fresh produce in season only, purchase canned fruits and vegetables without added sugar or salt. c) Cut foods into smaller pieces to reduce cost per ounce. d) Buy foods in bulk. e) Encourage alternative sources of protein. f) Support food preparation skill building and preparing food on a budget. |

(Continued)

**Table 3.** (Continued)

| Reference | Objective | Study Design & Methods | Target Population & Sample Size | Results | Recommendations for Care Providers |
|---|---|---|---|---|---|
| Lyles et al., 2013 [39] (USA) | Examine the relationship between food insecurity and A1C longitudinally. Examine secondary outcomes of self-reported diabetes self-efficacy and dietary intake of fruits and vegetables. | Secondary observational analysis of an intervention trial Analyzed baseline food insecurity in relation to A1C, self-efficacy, fruit/vegetable intake. | 665 low-income patients with diabetes, who received self-management support as part of larger diabetes education intervention in the United States. Participants were eligible if they received care at 1 of the 9 participating clinics, had A1C > 6.5%, spoke English, and did not have significant auditory, visual, or cognitive impairments. | Food insecure individuals had poorer A1C at baseline, but had greater improvements in A1C and self-efficacy following intervention, compared with food secure individuals. | a) Provide targeted self-management support to food insecure patients b) Do not assume that food insecure patients cannot improve dietary behaviours due to limited ability to afford healthy foods. |
| Seligman et al., 2018 [38] (USA) | To ascertain whether the provision of diabetes self-management education and diabetes appropriate food delivered in a food bank setting can help food insecure and diabetic clients achieve glycemic control | Randomized control trial 6 month intervention consisting of diabetes appropriate food (2x/month), referrals to health services, diabetes self-management education (DSME), and glucose monitoring. Primary outcome of interest was HbA1c levels at 6 months. | 568 adult food pantry clients with an HbA1c ≥ 7.5 in the United States. | Following the intervention, participants' food security (RR = 0.85), food stability (RR-0.77), and fruit and vegetable intake (RD = 0.34) increased. No observable changes were seen in self-management (depressive symptoms, diabetes distress, self-care, hypoglycemia, or self-efficacy) or HbA1c levels (RD = 0.24). | a) Food banks may be an optimal setting to distribute healthy foods appropriate for clients with diabetes to increase food security and intake of nutritious foods. b) Food banks often associate with the most vulnerable populations in society and as such are ideally positioned to partner with other community organizations to implement related interventions. |
| Seligman et al., 2015 [43] (USA) | Explore the feasibility of using food banks and their partner food pantries to provide diabetes support through a pilot intervention. | Pilot intervention Intervention provided clients with diabetes-appropriate food, blood glucose monitoring, primary care referral, and self-management support. Measures included A1C, diabetes self-efficacy, Medication Adherence Questionnaire, and food box satisfaction. | 687 food pantry clients with diabetes in three states in the United States over 6 months. | Improvements were seen in pre-post analyses of glycemic control (mean A1C decreased 8.11% to 7.96%). Among participants with baseline A1C ≥ 7.5%, A1C improved from 9.52% to 9.04%. Found significant improvements in fruit and vegetables intake, self-efficacy, diabetes distress, medication nonadherence, and trade-offs between buying food or medicine. | a) Consider a health promotion model for vulnerable populations through food banks and pantries. |
| Seligman et al., 2010 [15] (USA) | Assess whether food insecurity is associated with multiple indicators of diabetes self-management (self-efficacy, medication adherence, glucose-monitoring adherence, hypoglycemia, and glycemic control) among low-income adults with diabetes. | Cross-sectional A study conducted within a larger study examining the association between health literacy and cardiovascular disease. Interviews used six-item Food Security Survey Module and questions related to medication adherence, blood glucose testing, hypoglycemia, and trade-offs between food and medications. | 40 low-income adults aged ≥18 years with hypertension, on antihypertensive medication, seeking care at one of four safety net clinics in Chicago or Shreveport. Participants also had diagnosis of diabetes with one or more measures of A1C and current use of diabetes medications. | Food insecurity is a barrier to diabetes self-management and a risk factor for clinically significant hypoglycemia. Mean self-efficacy score was lower among food insecure than food secure participants (34.4 vs. 41.2, p = 0.02). FI participants reported poorer adherence to blood glucose monitoring (RR = 3.5, p = 0.008) and more hypoglycemia-related emergency- department visits (RR = 2.2, p = 0.007). Mean A1C was 9.2% among FI and 7.7% among FS participants (p = 0.08). | a) Screen low-income patients with diabetes for food insecurity to identify elevated risk of hypoglycemia and to tailor treatment decisions. b) Emphasize cost-neutral strategies in dietary counseling, such as reduced portion sizes rather than food substitutions. c) Use medications that carry lower risk of hypoglycemia (metformin and sulfonylureas). d) Adjust glycemic target upwards to mitigate elevated hypoglycemia risk associated with food insecurity. |

*(Continued)*

**Table 3.** (Continued)

| Reference | Objective | Study Design & Methods | Target Population & Sample Size | Results | Recommendations for Care Providers |
|---|---|---|---|---|---|
| Silverman et al., 2015 [28] (USA) | Evaluate the relationship between food security status and depression, diabetes distress, medication adherence, and glycemic control. Determine whether these factors can explain the relationship between food insecurity and glycemic control. | Secondary analysis of RCT data. Data obtained from Peer Support for Achieving Independence in Diabetes (Peer-AID). Measures include USDA 6-item Short Form Food Security Survey Module, PHQ-8, SF-12, Summary of Diabetes Self-Care Activities, and A1C. | 287 participants with poorly controlled (HbA1C $\geq$ 8.0) type 2 diabetes, household income < 250% of the federal poverty line, aged 30–70 recruited from three healthcare systems in Washington, USA | Individuals with food insecurity had greater odds of depression (OR 2.82–95% confidence interval [CI] 1.50–5.31, p = 0.001), diabetes distress (OR 2.32 [CI] 1.38–3.91, p = 0.002), and lower medication adherence (OR 1.96, [CI] 1.15–3.35, p = 0.01) compared with individuals who are food secure. Depression ($\beta$ = 0.55, p = 0.03) and diabetes distress ($\beta$ = 0.64, p = 0.03) are associated with higher mean A1C values. | a) Identify patients with food insecurity through screening to detect individuals with increased risk for poor health outcomes. b) Develop targeted interventions such as treating depression or addressing difficulties with medication adherence. |
| Soba et al., 2014 [45] (USA) | Implement an evidence-based food insecurity screening module for high-risk patients. | Pilot intervention Grey Literature Patient-centered approach to screen high-risk patients for food insecurity and appropriately tailor and manage care to improve outcomes. Measures included food insecurity screening rate and A1C. | 561 low-income adults $\geq$ 18 with type 2 diabetes and at high risk of food insecurity in the United States. | Rate of screening for food insecurity increased from baseline value of 0% to 82% after the 3-month implementation phase. 18% improvement in A1C, from > 7% to < 7% (p = 0.0001). | a) Provide a step-wise approach for care providers to follow when counseling food insecure patients with diabetes. b) Use patient education handouts re: food assistance and budget tips. |
| Thomas et al., 2018 [42] (Canada) | Evaluate the acceptability and feasibility of a food insecurity screening tool among patients with diabetes. | Pilot study 5 Canadian health care providers used the screening initiative, consisting of 3 questions and a corresponding care algorithm, for 2 weeks. | 33 patients $\geq$ 18 years old with type 2 diabetes were screened for food insecurity using the food insecurity screening questions and received diabetes management information from the care algorithm. 7 patients were interviewed regarding their experience with the screening initiative. 5 health care providers (4 dietitians and 1 nurse) were interviewed regarding the acceptability and feasibility of the screening initiative. | The food screening initiative provided patients with the opportunity to discuss food insecure circumstances. Overall, the initiative was found to be acceptable as the questions were simple to comprehend, did not affect patient's relationship with the care provider, and provided dietitians with pertinent information. Patient-provider familiarity increased patient's comfort during the screening as well. The initiative was also feasible as the care providers already screened for food insecurity in their practice but appreciated the systematic approach provided by the questions. Incorporation of the screening questions within an electronic medical report helped to remind providers to screen patients and allowed them to do so easily. | a) Care providers can use a systematic food insecurity screening tool that is incorporated into electronic medical records to more easily screen patients for food insecurity. b) Patients were comfortable discussing food insecurity with care providers and screening did not compromise rapport. |

*(Continued)*

**Table 3.** (Continued)

| Reference | Objective | Study Design & Methods | Target Population & Sample Size | Results | Recommendations for Care Providers |
|---|---|---|---|---|---|
| Vivian et al., 2014 [40] (USA) | Identify the self-care needs of adults with diabetes who experience food insecurity. | Cross-sectional A modified version of the Diabetes Knowledge Test was administered to measure general diabetes and insulin use knowledge. | 153 adults ≥ 18 with self-reported diabetes utilizing the St. Vincent de Paul Food Pantry in Wisconsin, USA. Participants had total household income below 185% of the federal poverty level. | Participants with post-secondary education or those who received diabetes education scored significantly higher on the diabetes knowledge test, compared with those with a high school education or less and those who did not receive diabetes education (p<0.05). Adults with type 1 diabetes had higher general and insulin use scores, compared with adults with type 2 diabetes, though scores were not statistically significant. | a) Screen all low-income patients with diabetes for food insecurity. b) Assess patients' gaps in diabetes knowledge and identify inappropriate self-care behaviours. c) Assessments should include medical history, demographics, cultural influences, health beliefs and attitudes, diabetes knowledge, diabetes self-management skills and behaviours, emotional response to diabetes, readiness to learn, literacy level, physical limitations, and family and financial support. d) Provide referrals to food resources, nutrition counseling that recognizes the challenges of food insecurity, smoking-cessation support, and appropriate medication management. |

into similar strategies or interventions, then placed under larger, theme-based headings. Development meetings were held with co-investigators and care providers who specialize in pediatric and adult diabetes care in Toronto, Canada to discuss these strategies for care. This information was later translated into an algorithm to guide clinical decision making to be published elsewhere.

## Results

From the scoping review, we compiled emerging practices to better support diabetes self-management among food insecure populations with diabetes. For our review, we only reported emerging practices that are not already commonplace in practice guidelines for general diabetes management [33]. Most were conducted in the United States, except for 8 studies from Canada. Only 3 of the 21 studies assessed diabetes self-management in food insecure pediatric populations [6,34,35]. The studies comprised of reviews [4,21,36,37], randomized control trials [38], secondary analyses [28,39], cross-sectional studies [5,6,15,27,40,41], pilot interventions [42,43], qualitative studies [25,35], commentaries [24,44], and grey literature [34,34].

The emerging practices for diabetes management among food insecure populations are organized according to interventions: food insecurity screening, nutrition counseling, improving glycemic control through medication management, building supportive care provider-patient communication and relationships, constructive coping, education, referring clients to food resources and supporting smoking cessation (see Table 4).

## Discussion

### Food Insecurity screening

The first step in addressing food insecurity among people with diabetes is identifying them. There is growing consensus about the necessity for routine food insecurity screening among adults and children with diabetes, conducted in a respectful and non-judgmental manner [5,6,15,21,24,25,27,28,34,40,44,45,46]. Vivian et al. recommend comprehensive assessments of food insecure adults with diabetes to identify knowledge gaps and harmful self-care behaviours that could impact patients' glycemic control [40]. Similarly, a study not in our review but conducted in low income patients, Pilkington et al., recommend learning about patients' life circumstances, exploring challenges to diabetes self-management, and helping patients access available resources [47]. Such assessments can enable care providers to tailor self-management plans, resulting in more realistic dietary advice and more appropriate medication regimens [40]. Although comprehensive assessment is time consuming, the information provided by patients following screening is rich and helpful for care providers [42]. Furthermore, these discussions can be spread out over several visits.

An unpublished thesis of a food insecurity screening initiative with 561 low-income adults with diabetes used two simple screening questions [45]. A treatment algorithm was then developed to guide care for patients identified as food insecure. The program increased the proportion of vulnerable patients with diabetes screened for food insecurity from 0% to 82%, and after 3 months there was a significant 18% reduction in the number of participants with hemoglobin A1C (A1C) levels above 7% [45]. Discussions regarding diabetes management on a budget (i.e. grocery shopping advice on healthy and affordable food), education on self-management when quantity and frequency of food intake were compromised, applicable information on local food assistance programs, and provision of nutrition information handouts to patients were instrumental to the intervention's success [45]. A pilot screening initiative reported by Thomas et al. examined the acceptability and feasibility of food insecurity screening among adults with diabetes in a community health center [42]. The initiative, which

**Table 4. Emerging practices identified.**

Food Insecurity Screening

- Screening patients and families for food insecurity is recommended as part of routine care [5,6,15,21,24,25,27,28,34,40,44,45]. Food security status should be assessed in an ongoing manner to provide most up-to-date information [25]
- A comprehensive assessment of patients' food security status helps to identify patients' psycho-social situation and allows care providers to tailor medical and dietary treatment plans to patients' circumstances [42]

Nutrition Counseling

- Registered dietitians can advise patients on ways to extend their budget and plan nutritious yet cost-effective meals to make self-management plans more realistic [25]
  - Encourage patients to eat out less and, purchase frozen or canned (with no added sugar or salt) fruits and vegetables when they are not in season [44]
  - Support patients to incorporate less costly protein sources into diets, such as legumes, eggs, and tofu [44]
  - Focus on reducing portion sizes of available foods (if appropriate) if patients are unable to make substitutions for healthier alternatives (may not be suitable for pediatric patients) [15,44,25]
- Encourage open conversations and reduce stigma associated with food insecurity by posting posters and resources that acknowledge the challenges of managing diabetes and eating healthfully [5]
- Support patients and their families to improve their food skills by showing patients how to prepare food and meals [44]

Improving Glycemic Control and Access to Medications

- Screen food insecure patients for occurrence and risk of hypoglycemia at every visit [44]
- Prescribe anti-hyperglycemic medications that are less likely to cause hypoglycemia (i.e. metformin, DPP-4 inhibitors, GLP-1s, and SGLT-2s) and consider increasing glycemic targets in adults and patient-specific glycemic targets in children; however, it should be noted that some of these medications are expensive and may not be covered by insurance [25,15]
- Tailor medical management to prevent hypoglycemia in the absence of food:
  - Prescribe longer acting insulin analogs or insulin degludec to prevent hypoglycemia when food supply is unpredictable, if feasible and affordable [25,44]
  - Prescribe more flexible insulin regimens to allow patients to omit doses in the absence of food [44]
  - Recommend scheduling medications with meals, rather than by time of day [25,44]
  - Instruct patients on how to alter diabetes medication to match food intake [36]

Improving care provider-patient communication and relationship

- Explain laboratory and exam results clearly and without judgement [25]
- Involve patients in the decision-making process [25]
- Develop strong rapport with patients by exhibiting compassion and empathy, particularly concerning food insecurity [42]

Coping Strategies

- Assess patients' coping strategies and address symptoms of diabetes distress, poor stress management and, poor coping [28]
- Refer patients to counseling services, if appropriate [28]

Referral to Community Resources

- Deliver health care self-management support services related to food, income and housing, such as prescription food programs and literacy appropriate educational material, if available [6,27,38,39]
- Provide patients with a list of local resources (affordable grocery stores, markets, meal delivery services, and organizations that provide free or low-cost meal), informing them about local community kitchens, education- and skill-building programs that help individuals utilize food resources more efficiently, and facilitate access to those resources by providing patients with contact information [25,27,36,44]

Smoking Cessation

- Provide smoking cessation support to potentially increase available funds for food as opposed to cigarettes, if appropriate [44]

included three screening questions and a care algorithm, demonstrated that patients are willing to share their experiences of food insecurity, despite acknowledging the sensitivity of the topic. Furthermore, screening elicited valuable information from patients that directed care providers' tailoring of treatment and care to best support food insecure patients [42]. Similarly, a food insecurity screening initiative implemented in a pediatric diabetes clinic revealed that most families were comfortable sharing food insecurity circumstances with care providers and appreciated the additional resources and care that accompanied a positive screening result [34]. These pilot studies provide promising examples of the acceptability and feasibility of food insecurity screening in routine diabetes care and its potential to improve glycemic control.

## Nutrition counseling

The role of dietary counseling for individuals with diabetes in guiding their purchase and preparation of healthy foods is well documented [27,41,48]. Regardless of food security status, maintaining a therapeutic diet is one of the more difficult elements of diabetes management [49]. However, the costs associated with such diets, as well as lack of access to cooking equipment, such as stoves, pose significant barriers to those who are food insecure [8]. In addition to findings from our review, challenges have been reported with portion size control and consumption of unbalanced meals with high starch and low vegetable content for low income individuals [49]. Those who are food insecure often resort to low-cost, energy-dense foods that contain refined carbohydrates, added sugars, and added fats [15,39,44].

Food insecure households may benefit from specific and tailored advice on extending their budgets, planning healthy-yet-affordable meals, and learning how to use their available resources more effectively [13,8]. Hence, referring food insecure patients with diabetes to registered dietitians is recommended [25], as they can support patients in following therapeutic diets on low budgets. For example, rather than recommending that patients choose healthier, more expensive brown rice, it may be more effective to focus on reduced portion sizes of more affordable white rice [15,44]. However, such practices may not be appropriate for children, as they have specific nutritional requirements during vital growth periods. Working within the budgets and foods accessible to individuals or families will make dietitians' recommendations more cost neutral and realistic [25]. Cost-saving strategies include eating out less, buying out-of-season frozen or canned fruits and vegetables with no added salt or sugar, and supporting individuals to eat cheaper proteins, such as beans and lentils, by showing clients how to cook these proteins [44]. Print resources that acknowledge the challenges of healthy eating on low budgets may reduce stigma and open conversations with clinicians about food insecurity [5]. Encouraging parents to enroll children in subsidized-school-meal programs can also relieve pressure on family budgets [44].

## Improving glycemic control and access to medications

Glycemic control depends, in-part, on quality of food choices and medication adherence [15]. Lopez & Seligman recommend screening food insecure patients for hypoglycemia at every visit [44] (e.g., asking patients about hypoglycemia symptoms or any blood glucose values below 4 mmol/L). If food insecure patients skip meals, clinicians can reduce their hypoglycemia risk by prescribing medications less likely to cause hypoglycemia (e.g., metformin, GLP-1s, DPP-4 inhibitors, SGLT-2s) and scheduling medication-taking with meals, not time of day [25,44]. Prescriptions for longer-acting insulin analogs, or insulin degludec can prevent hypoglycemia during unpredictable food supply periods [44]. More intensive diabetes-management methods (e.g., multiple daily basal- and bolus-insulin injections) can be modified to omit doses without food. Essien et al. propose loosening medical management restrictions to

prevent hypoglycemia; for example, teaching patients to alter medications when dietary intake is low or absent [36] and raising adult glycemic targets to reduce hypoglycemia risk [15]. Patient-specific glycemic targets may be more appropriate for children. Care plans for food insecure patients should further consider their medical and drug-formulary coverage to decrease expenditure (e.g. prescribing medications covered by social-assistance drug benefits or compassionate drug assistance programs) [46]. Essien et al. highlight the importance of balancing hypo- and hyperglycemia risk, particularly at times when patients are likely have used up their monthly income [36].

### Improving care provider-patient communication and relationships

Food insecurity, a sensitive topic, must be addressed without judgement in terms of how households prioritize their spending [42]. Genuine, empathetic, and non-judgmental, care is critical in supporting diabetes self-management [25] in this population. To enable food insecurity disclosure, positive patient-care provider communication and relationships are needed. A strong rapport with care providers has been shown to increase patients' comfort in answering food security screening questions [42]. Care providers are most helpful when they communicate clearly, elicit patient concerns, explain laboratory results and exam findings, and involve patients in decision-making [25,50]. Furthermore, it has been reported that socially disadvantaged individuals (i.e. individuals who are racialized or of a low socio-economic status) with diabetes benefit from frequent contact [35] of at least 10 hours in duration with nutrition educators over 6 months [51,52], allowing them to discuss challenges in following self-management recommendations.

### Coping strategies

Food insecure adults in general report more frequent stress, anxiety, and depression associated with a sense of powerlessness [28]. Higher stress levels may lead to decreased diabetes self-care and diabetes distress, both of which have been shown to be associated with suboptimal glucose control [28]. Poorly coping individuals may be less likely to adhere to medication regimes because of stress and/or financial strain, and their distress may increase when blood glucose levels rise, contributing to a vicious cycle of suboptimal blood glucose control. It is therefore important to assess patients' coping strategies and stress management skills [28,53,54] and to treat signs of stress or poor coping (e.g. depression, burnout, frustration, concern, apathy) to remediate food insecurity effects on glycemic control and potentially refer them to counseling services [28]. More attention should be given to stress management as a point of intervention to improve health outcomes of food insecure individuals with diabetes given the elucidated pathway between high stress levels, decreased diabetes self-care, and poor glycemic control [22].

### Referral to community resources

Referrals to sources of inexpensive food for food insecure households among people with diabetes support diabetes self-management [6,27]. These referrals may be to food banks/pantries, social assistance programs, affordable grocery stores, meal delivery services, organizations providing free or low-cost meals, and other supplemental food programs [25,27,36,44]. Connecting households with such government and community programs not only enables food access but may help ease other competing budget demands [36]. However, in a very recent published study not included in our review described how solely informing patients of services is insufficient and results in low usage rates. Instead, active enrolment on-site (i.e. in a clinic) that is

straightforward and facilitated by staff has shown to be more effective in achieving higher service usage rates [17]. Service providers must also ensure non-judgmental interactions with patients, as food insecurity often elicits a sense of shame and loss of dignity which can result in a reluctance to use food assistance services [55]. As such, suitable and appropriate referrals are necessary to help patients receive optimal care [25].

Ippolito et al. examined the association between food security and diabetes self-management among food pantry clients and concluded that food insecure individuals are less likely to access clinical care as frequently as their food secure counterparts [41]. As a result, food banks have also begun to partner with registered dietitians, delivering diabetes self-management support and glucose monitoring [41,38]. Offering healthcare support services through food banks and pantries reaches marginalized populations and addresses care gaps they may periodically experience [2,41,38,37]. An intervention study conducted by Seligman et al. demonstrated the effectiveness of providing diabetes-appropriate foods and self-management education in food banks to increase access to diabetes-appropriate foods and consumption of fruits and vegetables and reduce food insecurity [38]. Using a diabetes educational guide suited to all literacy levels, Lyles et al.'s diabetes self-management education intervention supports development of patient-centered self-management plans [39]. This approach resulted in significantly lower A1C and greater self-efficacy among food insecure individuals, compared with food secure individuals, even when the intervention is not focused directly on food insecurity [39]. Findings suggest targeted self-management educational support can improve clinical and behavioural outcomes among food insecure patients with diabetes.

Furthermore, programs that allow healthcare providers to write food prescriptions (i.e., coupons for healthy foods redeemable at participating retailers) can also improve diet quality [41]. Food prescription programs, beginning to emerge in America, legitimize the need for nutritionally adequate foods required for therapeutic diets. Additionally, having food prescriptions for healthy foods (i.e. fruits and vegetables, whole grains, seafoods, and nuts and seeds) being covered by Medicaid/Medicare has been shown in a simulation study by Lee et al. to potentially reduces formal healthcare expenditure by $100.2 billion and prevent 0.12 million diabetes diagnoses over the lifetime of those currently under the coverage of Medicaid/Medicare [56]. This finding suggests the cost-effectiveness, favourability, and need for healthy food prescriptions [56]. Similarly, two studies showed the effectiveness of food assistance programs in improving health outcomes. A food assistance intervention by Palar et al. provided meals that fully satisfied caloric and nutritional requirements to low-income participants with HIV or diabetes and observed an increase in fruit and vegetable consumption, and reduced frequency of sugar and fat intake, food insecurity, diabetes distress, and depressive symptoms [57]. There was also a reduction in participants forgoing food for healthcare and medication [57]. Cavanaugh et al. also showed that food prescription programs reduced BMI in a low-income population with diabetes [58]. These food prescriptions may also reduce stigma associated with households' need to use food assistance programs.

## Smoking cessation

Approximately a third of adults with diabetes are cigarette smokers, and those who are food insecure are twice as likely to smoke [2]. Asking about smoking habits in clinical assessments and non-judgmentally supporting patients to quit smoking, could help to alleviate budgetary constraints [40,44]. Care providers are urged to inform patients with diabetes about the risks of smoking and benefits of quitting [44]. When patients express interest in quitting, clinicians should provide information about public health programs that offer free smoking cessation counseling and non-prescription nicotine replacement therapy [44]. Some community health

centers may have respiratory therapists who can counsel referred patients on smoking cessation [44]. By reducing smoking, clinicians can support patients in decreasing expenditure on cigarettes and use the resulting additional funds on healthy foods as there is a dose-response relationship between increased cigarette spending and lower food spending [44]. Additionally, smoking has been linked to increased insulin resistance; thus, smoking cessation can improve glycemic control and prevent vascular complications that are common in those with diabetes [59].

## Limitations and future research

A major limitation of this scoping review is the dearth of research on interventions supporting food insecure people with diabetes, especially children, and no information was available for gestational diabetes and pre-diabetes. Additionally, although the search was not specific to North American studies, all eligible studies were from Canada or the United States. The lack of identified studies outside North America can be partly attributed to the English language inclusion criteria. As such, our results do not inform a global perspective. More studies with evaluative components are also needed to better direct clinical practice. Many of the interventions we reviewed do not measure clinical outcomes maintenance after participation ends. Given the above-mentioned limitations and until further evidence is available, our recommendations describe emerging practices, rather than inform practice guidelines. Research is needed to evaluate the effectiveness of these interventions on short- and long-term diabetes-related health outcomes.

## Conclusion

Clinicians can adopt several strategies to better support diabetes self-management among food insecure populations. Routine household food insecurity screening is a logical first step, followed by tailoring of diabetes management plans and interventions via medication management, community referrals, assessing coping strategies, supportive care provider-patient relationships, and smoking cessation. However, given the lack of studies, especially outside North America and in populations with gestational and prediabetes, more studies that evaluate the effectiveness of the identified emerging practices are needed to better inform health care providers and provide a global perspective.

## Supporting information

**S1 File. PRISMA checklist.**
(DOCX)

## Acknowledgments

The authors would like to thank the Lawson Foundation, Ryerson University Health Fund, the Division of Endocrinology at the Hospital for Sick Children, and the South Riverdale Community Health Centre for their support of this research. We thank the student volunteers who helped throughout the literature-review process.

## Author Contributions

**Conceptualization:** Enza Gucciardi, Jessica Wegener, Vanita Pais.

**Formal analysis:** Adalia Yang, Katharine Cohen-Olivenstein, Brittany Parmentier.

**Funding acquisition:** Enza Gucciardi.

**Methodology:** Enza Gucciardi.

**Supervision:** Enza Gucciardi.

**Writing – original draft:** Enza Gucciardi, Adalia Yang, Katharine Cohen-Olivenstein, Brittany Parmentier.

**Writing – review & editing:** Enza Gucciardi, Adalia Yang, Katharine Cohen-Olivenstein, Brittany Parmentier, Jessica Wegener, Vanita Pais.

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
