## [Decision Letter · Decision Letter 0]

13 Aug 2019

PONE-D-19-15760

Emerging practices supporting diabetes self-management among food-insecure adults and families: A scoping review

PLOS ONE

Dear Dr. Gucciardi,

Thank you for submitting your manuscript to PLOS ONE. After careful consideration, we feel that it has merit but does not fully meet PLOS ONE’s publication criteria as it currently stands. Therefore, we invite you to submit a revised version of the manuscript that addresses the points raised during the review process.

We would appreciate receiving your revised manuscript by Sep 27 2019 11:59PM. To enhance the reproducibility of your results, we recommend that if applicable you deposit your laboratory protocols in protocols.io, where a protocol can be assigned its own identifier (DOI) such that it can be cited independently in the future. For instructions see: http://journals.plos.org/plosone/s/submission-guidelines#loc-laboratory-protocols

We look forward to receiving your revised manuscript.

Kind regards,

David Alejandro González-Chica, Ph.D., M.D.

Academic Editor

PLOS ONE

Journal Requirements:

Reviewers' comments:

Reviewer's Responses to Questions

**Comments to the Author**

1. Is the manuscript technically sound, and do the data support the conclusions?

Reviewer #1: Yes

Reviewer #2: Partly

Reviewer #3: Yes

2. Has the statistical analysis been performed appropriately and rigorously? 

Reviewer #1: N/A

Reviewer #2: N/A

Reviewer #3: N/A

3. Have the authors made all data underlying the findings in their manuscript fully available?

Reviewer #1: Yes

Reviewer #2: Yes

Reviewer #3: Yes

4. Is the manuscript presented in an intelligible fashion and written in standard English?

Reviewer #1: Yes

Reviewer #2: Yes

Reviewer #3: Yes

5. Review Comments to the Author

Reviewer #1: Preamble

The paper presents a scoping review on emerging practices supporting diabetes self-management among food insecure adults and families and relied on different types of studies. In total, 21 articles were included in the review and discussed with respect to a number of inclusion criteria. The review methodology meets expectations for scoping reviews, most notable the use of Preferred Reporting Items for Systematic Review and Meta-Analysis (PRISMA) extension for scoping review. The paper which addresses an important topic- considering how health professionals could support diabetes self-management among food insecure patients and families is very timely and relevant to current healthcare.

Points for the authors to consider:

Introduction

1. Although the background provides a clear rationale for the study, the literature and summary on the prevalence of food insecurity was mainly on studies from North America and Canada. Whereas in the methodology, the data search include studies from any location or setting. Please, amend this discrepancies by including prevalence data outside the aforementioned specific locations and if possible global data should be included (if available in the literature).

2. The statement: ‘‘The likelihood of food insecurity increases by 4% with each year of age earlier someone is diagnosed with diabetes [2].’’ is not clear. Please rephrase for a better clarity.

3. The last statement in the introduction: ‘‘Policy interventions to address poverty, employment, housing, and income, are fundamental in alleviating food insecurity [4]’’ stands aloof. I can’t see the necessity for this statement at the end of the introduction. It needs further explanation and proper integration into the paragraph. If not, you might consider a total removal.

Methodology

Data Sources and Search strategy:

1. PRISMA extension for scoping review checklist requires a citation.

Study Selection:

1. Please include start and end dates of the articles search.

2. In your introduction, you defined ‘‘emerging practices as new and innovative interventions which holds promises for best practices……. The above statement does not seems to be taken into considerations under the explanations provided in the study selection. It seems, you included ANY (old, new, innovative or not) article with strategies or interventions for insecure populations with diabetes which could be operationalised by care providers’’. Please clarify these inconsistencies.

Fig 1. PRISMA chart outlining data study selection process.

1. Some of the calculations does not sum up. e.g 3066 + 7 cannot give 3078. Please make corrections accordingly.

Table 1 and 2

1. Please include study location. Although, locations were included in some of the studies but it is lacking in majority, especially in studies whose design are not reviews or commentary.

Results.

1. It seems the reviewed articles were eventually streamlined to studies carried out in the USA or Canada alone. Although, this was not included under the eligibility nor key word search. Since, this is a review, as a standard, a reader would expect the study to be a bit generalizable. I assume, the review focus was not on these locations alone but there were limited literatures available from other countries/continents. Therefore, I would strongly suggest that the authors state this very clearly in the Limitations. But if the study focus was on the aforementioned locations, I advise, it should then reflect in the article topic, as this would alert the reader upfront about the content of the review.

2. The article which was referenced with citation number ‘‘55’’[Segman etal., 2015] under table 2 is missing in the study design summary. It was not grouped under pilot interventions nor grey literature as stated in the table.

Discussion

The discussion appear well thought-out and explanatory.

Reviewer #2: Thanks for your work. The study is very relevant I think. One of the most important drawbacks of this study is that the it focusses on North-American countries and does not take a more global perspective (at least that is what I assume). Please include this in your limitation section.

More clarity on the following points would benefit the manuscript:

Introduction

Please state that you focus on North-American populations. This should be clearer mentioned in the introduction and certainly in the methods section. In case the authors attempted to write this review for global populations (which is not the case I assume), this would not be an appropriate study.

Authors focused on self-management support by health care providers (please specify in the objectives). This formulation is a bit unfortunate because several of the recommendations are not related to self-management: e.g. screening, prescribe a certain type of medication, monitor coping strategies, etc. In this case, a better formulation would be something in the trend of: “person-centered” approach/practices by health providers, eventually with the aim to support “nutrition related self-management”. The purpose is to indicate that the practices relate to actions undertaken by the provider, rather than by the patient.

Methods

Please state that you focus on North-American populations (which I assume).

When was the start of inclusion of papers, please specify?

Authors claim that there are no guidelines for scoping reviews. What about this:

https://wiki.joannabriggs.org/display/MANUAL/11.3.7.1+Search+strategy

Why is there a study of 2019 included; Vitale et al?

You define interventions as having “addressed diabetes self-management in a food insecure population”. However, I don’t agree with what you call self-management support (see comment in the introduction). An extra search including terms like “tailored” support, “person-centered” care etc. is warranted.

The search terms mentioned in the methods do not correspond to the ones in the search strings in table 4. Please be systematic.

The authors did an important job: a substantial part of the literature was screened and different search methods were used. However, a quick scroll through the literature resulted in the identification of the following papers:

DECKER, DOMINIC, and MARY FLYNN. "Food Insecurity and Chronic Disease: Addressing Food Access as a Healthcare Issue." Rhode Island Medical Journal 101.4 (2018).

Goddu, Anna P., et al. "Food Rx: a community–university partnership to prescribe healthy eating on the South Side of Chicago." Journal of prevention & intervention in the community 43.2 (2015): 148-162.

It is not clear why these papers were not identified (maybe they did no respond to the inclusion criteria?).

Also, there is not sufficient information provided to repeat the analysis; this may be included in supplementary files.

Results

The authors state: “some recommendations for adults with diabetes can be

adopted for children”

This is quite vague and the evidence to support this claim is lacking.

Discussion

Although interesting, the discussion section is a more detailed presentation of the results, which I would personally position under ‘results’. I am missing a critical discussion of the results and of the review methods and processes.

It is also unclear in some parts of the discussion if the authors rely on the information from the studies identified in their review, or on other studies and why. This should be better documented if they would keep the current structure (which I don’t recommend).

Some references in the discussion refer to other contexts than the focus of the review papers, this is very confusing and not relevant.

For instance: the reference to this study in South-Africa seems not appropriate since the review is focused on studies in North-America.

“Challenges have been reported with portion size control and consumption of unbalanced meals with high starch and low vegetable content for low income individuals”

Muchiri JW, Gericke GJ, Rheeder P. Needs and preferences for nutrition education of type 2

diabetic adults in a resource-limited setting in South Africa. Health SA Gesondheid

[Internet]. 2012;17(1). Available from: http://www.scopus.com/inward/record.url?eid=2-

s2.0-84877700781&partnerID=40&md5=51c092f32e25269fa8abdeed29eaf477

or this reference about Scotland:

Douglas F, MacKenzie F, Ejebu O, Whybrow S, Garcia AL, McKenzie L, et al. “A lot of

people are struggling privately. They don’t know where to go or they’re not sure of what to

do”: Frontline Service Provider Perspective of the Nature of Household Food Insecurity in

Scotland. Int J Environ Res Public Health. 2018;15:2737. doi:10.3390/ijerph15122738

The way some sentences are states are not very ‘person-centered’. For instance:

“Care providers are urged to inform patients with diabetes about the risks of smoking and benefits of quitting” or

“Clinicians should provide information about public health programs that offer free smoking cessation counseling and non-prescription nicotine replacement therapy.”

This sentence goes against the theory of Prochaska and Diclemente which takes into account the different stages of contemplation. If the aim is to promote self-management, practices should be backed-up by behavior & motivational theory.

The authors refer to a need for studies with “robust designs” in their limitations. This is a very confusing term.

Conclusion

The conclusion is poor and does not seem to correspond to the studies identified by the authors.

The sentence: “The adverse impact of food insecurity on diabetes self-management is well documented.” This may be a good sentence for the introduction, but this was not the aim of the study.

The following sentence: “Policies targeting underlying causes of food insecurity and poverty are desperately needed to improve overall health and quality of life” was not a result of the aimed review and should not be included.

The sentence: “Although there were few significant differences in recommendations for supporting children versus adults”

The study concluded that there was very limited evidence for children, how can the authors make this statement then?

Finally , the main results are repeated in the conclusion (this is the fourth time the authors repeat these results). The conclusion should entail a more general appreciation of the study findings, limitations and eventual future research. Some key findings may be repeated. Please take similar studies as an example.

Reviewer #3: Summary of review and Impression

This paper set out to examine emerging strategies and interventions presently utilized to improve diabetes self-management among food insecure populations. Using a scoping review and a PICOTS framework, the authors reviewed 21 articles meeting their set criteria.

The study topic is relevant and research area useful for control of diabetes. The overall process of the study was good as it followed identifiable scoping review methodology. The manuscript was well written with few specific areas that will require revision (see below). However, I found the conclusions as they were presented in the abstract as well as the body of the study not a strong reflection of the stated purpose of the study (see below).

Overall, the study presents important findings relevant to current topical issues related to diabetes management and will be good for publication if relevant revisions are made. My observation though is that this study does not seem to be significantly different from the review by Gucciardi et al, 2014 except for the title and purpose.

Areas of suggested improvements

Minor

Abstract

1. Study selection will need to be stated more clearly. The last sentence,” Of articles that fit the inclusion criteria, 21 were selected for review” …The sentence implies that some articles that met the inclusion criteria were not selected for review which will be incorrect. Were articles excluded after meeting the inclusion criteria?

2. The abstract presentation varies from the recommended PRISMA reporting structure ie background, objectives, eligibility criteria, sources of evidence, charting methods, results, and conclusions that relate to the review questions and objectives. Some sub-sections included in the abstract are not necessary ie limitation, study selection. I suggest another a revision of some of the components.

3. In the sub-section called “data synthesis”, it appears the study was referring to “result/finding”. The use of appropriate sub-title “results/findings” might be considered for the audience to follow as synthesis may suggest method rather than result.

4. The limitation needs to be improved on for more clarity. If the author says that, ” There are limited evaluations targeting food insecure individuals with diabetes”, does it mean they found few articles on the subject of food insecurity and diabetes management? If so, it will make for better reading if it is stated so and explain why it is a limitation for their study.

5. In the conclusion, the authors state that, “Food insecurity screening and subsequent tailoring of realistic diabetes management plans for food insecure patients may improve their observance of recommendations and glycemic control”, while this may be true, it would have been better if the conclusion focused on presenting the emerging practices that was found in the review (as it did later in the main body of the manuscript). It may be better to simply highlight what the review found as emerging practices supporting diabetes self-management in food insecure population without speculating on their effectiveness since the evidence of effectiveness had not been presented. The second sentence in the conclusion will then be well placed.

Main Body

1. In using PICOTS framework for article selection, the study was not clear on the difference between what it stated as the intervention (“practices & strategies”) and outcome (“practices”). The study objectives suggested interest in finding interventions rather than outcomes. Does the outcome in this PICOT refer to outcome of the articles that was reviewed or the outcome of this study? Seems both are confused here. This explanation of PICOTS’ use in the study needs to be refined or better explained.

2. The search conducted until the end of November 2018, retrieved 3066 articles (Fig 1)

[29] “ – Not clear why the citation is included here.

3. “Twenty-one articles were selected for review based on the inclusion criteria”. May be better to move this sentence to after, "acceptable articles were reviewed in full..."

4. “Full text-articles were necessary to be included in this review”. This needs some clarity and could be reworded.

5. In the result section, the statement, “Although research about families of children with diabetes is sparse, some recommendations for adults with diabetes can be adopted for children, appears to be more conclusion and discussion than result. Might be better to move it to discussion or conclusion.

6. I am not sure why the discussion section was presented in sub-titles. It may make for better reading if the subtitles are removed and the discussion given a good flow with a paragraph discussing each point stated in the present subtitles.

7. The conclusion seems lengthy, making it difficult to follow key take away from the study which was left till the last sentence. It might improve the article if more concise conclusion relevant to the objective of the article is written at the beginning and other less relevant sentences written later of left out entirely. The last sentence has the most essential conclusion (Also refers to the study aim) and should be the focus of the whole section.

6. PLOS authors have the option to publish the peer review history of their article (what does this mean?). If published, this will include your full peer review and any attached files.

Reviewer #1: Yes: Mary Damilola Adu

Reviewer #2: No

Reviewer #3: Yes: Bonaventure Amandi Egbujie

---

## [Author Response · Author response to Decision Letter 0]

10 Sep 2019

We would like to thank the reviewers for their constructive feedback and an opportunity to revise and resubmit our manuscript. Please find below our responses to the reviewers’ comments. Thank you.

Review Comments to the Author

Reviewer #1: Preamble

The paper presents a scoping review on emerging practices supporting diabetes self-management among food insecure adults and families and relied on different types of studies. In total, 21 articles were included in the review and discussed with respect to a number of inclusion criteria. The review methodology meets expectations for scoping reviews, most notable the use of Preferred Reporting Items for Systematic Review and Meta-Analysis (PRISMA) extension for scoping review. The paper which addresses an important topic- considering how health professionals could support diabetes self-management among food insecure patients and families is very timely and relevant to current healthcare.

Points for the authors to consider:

Introduction

1. Although the background provides a clear rationale for the study, the literature and summary on the prevalence of food insecurity was mainly on studies from North America and Canada. Whereas in the methodology, the data search include studies from any location or setting. Please, amend this discrepancies by including prevalence data outside the aforementioned specific locations and if possible global data should be included (if available in the literature).

Response: Prevalence data and literature where food insecurity was noted as a potential risk factor for diabetes was included in the introduction of our manuscript. Please see page 3, 2nd paragraph in blue text. Please see added text below:

A few studies outside of North America have identified a higher prevalence of food insecurity among those with diabetes or have identified food insecurity as a risk factor for poorer diabetes management. For instance, a study in Iran showed that those who were food insecure were 2.8 times more likely to have diabetes than those who were food secure [10]. In Kenya, food insecure individuals with diabetes were more likely to be on insulin or have had been on insulin compared to their food secure counterparts [11]. Another study reported severely food insecure Jordanians with diabetes had a higher body mass index (BMI) despite having a lower caloric intake than food secure or mildly food insecure individuals with diabetes [12].

2. The statement: ‘‘The likelihood of food insecurity increases by 4% with each year of age earlier someone is diagnosed with diabetes [2].’’ is not clear. Please rephrase for a better clarity.

Response: Thank you, we have re-worded the sentence for greater clarity. “The likelihood of food insecurity increases by 4% with every year earlier an individual is diagnosed with diabetes.” Please see page 3, 1st paragraph. 

3. The last statement in the introduction: ‘‘Policy interventions to address poverty, employment, housing, and income, are fundamental in alleviating food insecurity [4]’’ stands aloof. I can’t see the necessity for this statement at the end of the introduction. It needs further explanation and proper integration into the paragraph. If not, you might consider a total removal.

Response: We have deleted this sentence from the introduction as suggested. 

Methodology

Data Sources and Search strategy:

1. PRISMA extension for scoping review checklist requires a citation.

Response: We have the PRISMA extension checklist cited on page 6, 3rd paragraph.

Study Selection:

1. Please include start and end dates of the articles search.

Response: We include all articles up until November 2018. As such, there was no start date for our inclusion of articles. The sentence has been modified for greater clarity. See below:

“The search conducted for all dates up to November 2018, retrieved 3066 articles (Fig 1).”

2. In your introduction, you defined ‘‘emerging practices as new and innovative interventions which holds promises for best practices……. The above statement does not seem to be taken into considerations under the explanations provided in the study selection. It seems, you included ANY (old, new, innovative or not) article with strategies or interventions for insecure populations with diabetes which could be operationalized by care providers’’. Please clarify these inconsistencies.

Response: As suggested by the reviewer we have clarified the definition of emerging practices by adding the blue text in our manuscript. Please see page 6, 1st paragraph and below:

This paper will define emerging practices as recommendations, practices, strategies or “interventions that are new, innovative and which hold promise based on some level of evidence of effectiveness or change that is not research-based and/or sufficient to be deemed a ‘promising’ or ‘best’ practice” yet [29]. As such, practices that are currently in use but have yet to be substantially evaluated have been included. Emerging practices must also be based on “protocols, standards, or preferred practice patterns that [may] lead to effective – health outcomes” [30]. 

Fig 1. PRISMA chart outlining data study selection process.

1. Some of the calculations does not sum up. e.g 3066 + 7 cannot give 3078. Please make corrections accordingly.

Response: Thank you for catching this error. The PRISMA chart has been revised and corrected. 

Table 1 and 2

1. Please include study location. Although, locations were included in some of the studies but it is lacking in majority, especially in studies whose design are not reviews or commentary.

Results.

Response: Locations for all studies that are not reviews or commentaries have been added. Authors of reviews and commentaries were located either in Canada or the U.S. Locations for all review papers have been added in Table 1 and Table 2. 

1. It seems the reviewed articles were eventually streamlined to studies carried out in the USA or Canada alone. Although, this was not included under the eligibility nor key word search. Since, this is a review, as a standard, a reader would expect the study to be a bit generalizable. I assume, the review focus was not on these locations alone but there were limited literatures available from other countries/continents. Therefore, I would strongly suggest that the authors state this very clearly in the Limitations. But if the study focus was on the aforementioned locations, I advise, it should then reflect in the article topic, as this would alert the reader upfront about the content of the review.

Response: To address this comment we have added this text below to our limitation section. Please see page 35, 1st paragraph.

“Although the search was not specific to North American studies, eligible studies based on inclusion and exclusion criteria were from Canada or the United States. As such, our results do not inform a global perspective and hence more research is needed from outside of North America regarding strategies that may better support diabetes self-management among those challenged by food insecurity. “ 

2. The article which was referenced with citation number ‘‘55’’[Segman etal., 2015] under table 2 is missing in the study design summary. It was not grouped under pilot interventions nor grey literature as stated in the table.

Response: This has been corrected in the study design summary. Please keep in mind the reference numbers have changed due to the inclusion of additional articles to support the new global data paragraph added in the introduction. The article by Seligman et al. 2015 is now under reference number 59. 

Discussion

The discussion appear well thought-out and explanatory.

Reviewer #2: Thanks for your work. The study is very relevant I think. One of the most important drawbacks of this study is that the it focusses on North-American countries and does not take a more global perspective (at least that is what I assume). Please include this in your limitation section.

Response: During our literature search, we did not screen out any countries. Given our inclusion and exclusion criteria particularly, a strategy, intervention or practice was needed to be described or recommended for inclusion into our review. As such, only North American studies were found eligible based on our definition of emerging practices. Please refer to our screening criteria. We have noted this as a limitation of the current body of evidence in our limitation. Please see page 35, 1st paragraph

Excerpt from limitations section: “Additionally, although the search was not specific to North American studies, eligible studies based on inclusion and exclusion criteria were from Canada or the United States. As such, our results do not inform a global perspective and hence more research is needed from outside of North America regarding strategies that may better support diabetes self-management among those challenged by food insecurity.”

More clarity on the following points would benefit the manuscript:

Introduction

Please state that you focus on North-American populations. This should be clearer mentioned in the introduction and certainly in the methods section. In case the authors attempted to write this review for global populations (which is not the case I assume), this would not be an appropriate study.

Response: The objective of this paper was not to assess the prevalence of diabetes among those who are food insecure which would have included a few more international papers, which we included in our introduction (please see page 3, 2nd paragraph). However, we were looking for strategies that care providers can relay to patients or use to support those living with diabetes and challenged by food insecurity. Given this criterion we were only able to find North American studies. We have noted this in our limitation section. See page 35, 1st paragraph.

Authors focused on self-management support by health care providers (please specify in the objectives). This formulation is a bit unfortunate because several of the recommendations are not related to self-management: e.g. screening, prescribe a certain type of medication, monitor coping strategies, etc. In this case, a better formulation would be something in the trend of: “person-centered” approach/practices by health providers, eventually with the aim to support “nutrition related self-management”. 

Response: Our research objective was to identify recommendations, strategies, practices, or interventions that care providers can relay to patients or use to support those living with diabetes and challenged by food insecurity. We believe screening for food insecurity, medical management, monitoring of coping strategies, tailoring care, are all related to supporting self-management of diabetes among patients. We have revised our objectives to make this clearer see page 5, 2nd paragraph. Please see our clarified research objective below:

“The primary aim of this scoping review is to identify emerging practices that health care providers can relay to patients or operationalize to support diabetes self-management among food insecure populations.”

Methods

Please state that you focus on North-American populations (which I assume).

When was the start of inclusion of papers, please specify?

Response: There was no specific start of inclusion of our papers, as we reviewed all papers up until November 2018. This has been clarified for our readers as per below: 

“The search conducted for all dates up to November 2018, retrieved 3066 articles (Fig 1).”

Authors claim that there are no guidelines for scoping reviews. What about this:

https://wiki.joannabriggs.org/display/MANUAL/11.3.7.1+Search+strategy

Response: We have deleted the sentence that stated there were no guidelines for scoping reviews.

Why is there a study of 2019 included; Vitale et al?

Response: This article was grey literature known to the authors prior to the end of the search date (i.e. November 2018). It is now published and thus we are using the published date to help readers find the article if desired.

You define interventions as having “addressed diabetes self-management in a food insecure population”. However, I don’t agree with what you call self-management support (see comment in the introduction). An extra search including terms like “tailored” support, “person-centered” care etc. is warranted.

Response: Thank you for your comment. The focus of our paper was to look at strategies that can help patients to better self-manage their diabetes when challenged by food insecurity beyond what is already recommended by practice guidelines for the general diabetes population. Hence the keywords chosen ‘self-management or self-care’ we feel are most adequate. Ultimately, we are focusing on the management of diabetes. Thus, we wanted our search to be specific to self-management of diabetes. We recently ran searches with “tailored” and “person-centered” using the same timelines of our review and found no additional articles. 

The search terms mentioned in the methods do not correspond to the ones in the search strings in table 4. Please be systematic.

Response: Thank you for catching this. Please see below and see page 6-7 for revisions.

“Combinations of the following key words were used: diabetes, diabetes mellitus, type 1 diabetes, diabetes mellitus, type 1, type 2 diabetes, diabetes mellitus, type 2, gestational diabetes, gestational, prediabetes, prediabetic state, food security, food insecurity, food supply, cooking, food skills, education, patient education, health education, coping strategies, therapeutics, self-efficacy, diabetes management, self-management, self-care, low income, poverty, hunger, pediatric, newborn, infant, preschool child, child, adolescent, family characteristic, family, and household.”

The authors did an important job: a substantial part of the literature was screened and different search methods were used. However, a quick scroll through the literature resulted in the identification of the following papers:

DECKER, DOMINIC, and MARY FLYNN. "Food Insecurity and Chronic Disease: Addressing Food Access as a Healthcare Issue." Rhode Island Medical Journal 101.4 (2018).

Goddu, Anna P., et al. "Food Rx: a community–university partnership to prescribe healthy eating on the South Side of Chicago." Journal of prevention & intervention in the community 43.2 (2015): 148-162.

It is not clear why these papers were not identified (maybe they did no respond to the inclusion criteria?).

Response: These two papers were identified in our search but screened out. The fist paper does not focus on diabetes and does not identify any emerging practices. The second paper targets “underserved communities,” we specifically focused on the term “food insecure” to narrow our search so that emerging practices would be most relevant to our population of interest. Our search inclusion and exclusion criteria can be found on page 6, 2nd paragraph.

Also, there is not sufficient information provided to repeat the analysis; this may be included in supplementary files.

Response: An example of the search is provided in table 4 on page 42.

Results

The authors state: “some recommendations for adults with diabetes can be

adopted for children”

This is quite vague and the evidence to support this claim is lacking.

Response: We have removed this sentence from the paper. 

Discussion

Although interesting, the discussion section is a more detailed presentation of the results, which I would personally position under ‘results’. I am missing a critical discussion of the results and of the review methods and processes.

Response: This is a scoping review and describes what strategies are being recommended or used. We do describe how some studies lack thorough evaluation and are critical of our limitations in carrying out this review. Please see page 35, 1st paragraph. 

It is also unclear in some parts of the discussion if the authors rely on the information from the studies identified in their review, or on other studies and why. This should be better documented if they would keep the current structure (which I don’t recommend). Some references in the discussion refer to other contexts than the focus of the review papers, this is very confusing and not relevant.

Response: Our discussion summarizes the evidence but also pulls relevant literature that supports our findings. We have tried to be clearer in terms of describing the studies we reviewed versus the studies that provide support and extra context to our findings. We believe these references are relevant to our discussion. Please see discussion section on page 27.

For instance: the reference to this study in South-Africa seems not appropriate since the review is focused on studies in North-America.

“Challenges have been reported with portion size control and consumption of unbalanced meals with high starch and low vegetable content for low income individuals”

Response: Please see our response to the comment reviewer #2 made under the methods section above.

Muchiri JW, Gericke GJ, Rheeder P. Needs and preferences for nutrition education of type 2

diabetic adults in a resource-limited setting in South Africa. Health SA Gesondheid

[Internet]. 2012;17(1). Available from: http://www.scopus.com/inward/record.url?eid=2-

s2.0-84877700781&partnerID=40&md5=51c092f32e25269fa8abdeed29eaf477

or this reference about Scotland:

Douglas F, MacKenzie F, Ejebu O, Whybrow S, Garcia AL, McKenzie L, et al. “A lot of

people are struggling privately. They don’t know where to go or they’re not sure of what to

do”: Frontline Service Provider Perspective of the Nature of Household Food Insecurity in

Scotland. Int J Environ Res Public Health. 2018;15:2737. doi:10.3390/ijerph15122738

Response: Our review criteria focused on populations that have been defined as “food insecure”, but we have included other articles that have described “low-income or resource limited populations” in relation to food access in our discussion section to provide greater context in support of our findings. The second paper mentioned above was used in our discussion but was not eligible in our review as it included people with various chronic diseases, in addition to diabetes. We limited our review to those with diabetes. In our discussion we are situating our results in the broader literature, to further support and provide context to our review’s findings. Our review findings are described clearly in Table 3.

The way some sentences are states are not very ‘person-centered’. For instance:

“Care providers are urged to inform patients with diabetes about the risks of smoking and benefits of quitting” or

“Clinicians should provide information about public health programs that offer free smoking cessation counseling and non-prescription nicotine replacement therapy.”

This sentence goes against the theory of Prochaska and Diclemente which takes into account the different stages of contemplation. If the aim is to promote self-management, practices should be backed-up by behavior & motivational theory.

The authors refer to a need for studies with “robust designs” in their limitations. This is a very confusing term.

Response: Thank you, we have reworded the above sentence. 

“When patients express interest in quitting, clinicians should provide information about public health programs that offer free smoking cessation counseling and non-prescription nicotine replacement therapy”

We have also replaced “robust designs” with “evaluative components” in the limitations section for clarification as suggested by the reviewer. Please find this change on page 35, 1st paragraph. 

Conclusion

The conclusion is poor and does not seem to correspond to the studies identified by the authors.

The sentence: “The adverse impact of food insecurity on diabetes self-management is well documented.” This may be a good sentence for the introduction, but this was not the aim of the study.

The following sentence: “Policies targeting underlying causes of food insecurity and poverty are desperately needed to improve overall health and quality of life” was not a result of the aimed review and should not be included.

Response: We have revised our concluding paragraph based on comments from both reviewer 2 and 3. We have excluded the first and following sentence as suggested. 

The sentence: “Although there were few significant differences in recommendations for supporting children versus adults”. The study concluded that there was very limited evidence for children, how can the authors make this statement then?

Response: This was removed as suggested.

Finally, the main results are repeated in the conclusion (this is the fourth time the authors repeat these results). The conclusion should entail a more general appreciation of the study findings, limitations and eventual future research. Some key findings may be repeated. Please take similar studies as an example.

Response: Please see the revised conclusion on page 36, 1st paragraph. The limitations of the papers are discussed in a paragraph above the conclusions. We have revised the conclusion based on both reviewer 2 and 3 suggestions. As recommended, we have summarized our main findings and highlighted future direction for research.

Reviewer #3: Summary of review and Impression

This paper set out to examine emerging strategies and interventions presently utilized to improve diabetes self-management among food insecure populations. Using a scoping review and a PICOTS framework, the authors reviewed 21 articles meeting their set criteria.

The study topic is relevant and research area useful for control of diabetes. The overall process of the study was good as it followed identifiable scoping review methodology. The manuscript was well written with few specific areas that will require revision (see below). However, I found the conclusions as they were presented in the abstract as well as the body of the study not a strong reflection of the stated purpose of the study (see below).

Overall, the study presents important findings relevant to current topical issues related to diabetes management and will be good for publication if relevant revisions are made. My observation though is that this study does not seem to be significantly different from the review by Gucciardi et al, 2014 except for the title and purpose.

Response: The two papers are distinct in that in this current review we are focusing on strategies specifically for care providers to either convey to patients or to implement in their clinical practice to better support patients with diabetes. The previous review discussed the prevalence of food insecurity and diabetes globally, where data was available, the methodology used to assess food insecurity in the studies reviewed, and the various strategies to mitigate food insecurity broadly focusing on policy, community, and the healthcare system. 

Areas of suggested improvements

Minor

Abstract

1. Study selection will need to be stated more clearly. The last sentence,” Of articles that fit the inclusion criteria, 21 were selected for review” …The sentence implies that some articles that met the inclusion criteria were not selected for review which will be incorrect. Were articles excluded after meeting the inclusion criteria?

Response: Thank you for bringing this to our attention. The articles were not excluded after meeting inclusion criteria and the sentence has been re-worded to prevent confusion. Please see below:

“In total, 21 articles were reviewed.”

2. The abstract presentation varies from the recommended PRISMA reporting structure i.e., background, objectives, eligibility criteria, sources of evidence, charting methods, results, and conclusions that relate to the review questions and objectives. Some sub-sections included in the abstract are not necessary i.e., limitation, study selection. I suggest another a revision of some of the components.

Response: We have revised the abstract presentation so that it is aligned with the PRISMA reporting structure. Please see page 2 for abstract.

3. In the sub-section called “data synthesis”, it appears the study was referring to “result/finding”. The use of appropriate sub-title “results/findings” might be considered for the audience to follow as synthesis may suggest method rather than result.

Response: Thank you for your comment. We have revised according to your recommendations.

4. The limitation needs to be improved on for more clarity. If the author says that, ” There are limited evaluations targeting food insecure individuals with diabetes”, does it mean they found few articles on the subject of food insecurity and diabetes management? If so, it will make for better reading if it is stated so and explain why it is a limitation for their study.

Response: We have revised as recommended. See below:

Limitations was changed to: “Further research is needed in food insecure individuals with gestational diabetes, prediabetes, and outside of North America.” Please see page 3, 1st line.

5. In the conclusion, the authors state that, “Food insecurity screening and subsequent tailoring of realistic diabetes management plans for food insecure patients may improve their observance of recommendations and glycemic control”, while this may be true, it would have been better if the conclusion focused on presenting the emerging practices that was found in the review (as it did later in the main body of the manuscript). It may be better to simply highlight what the review found as emerging practices supporting diabetes self-management in food insecure population without speculating on their effectiveness since the evidence of effectiveness had not been presented. The second sentence in the conclusion will then be well placed.

Response: Thank you. The conclusion in the abstract has been revised to incorporate the reviewer’s suggestions. 

Main Body

1. In using PICOTS framework for article selection, the study was not clear on the difference between what it stated as the intervention (“practices & strategies”) and outcome (“practices”). The study objectives suggested interest in finding interventions rather than outcomes. Does the outcome in this PICOT refer to outcome of the articles that was reviewed or the outcome of this study? Seems both are confused here. This explanation of PICOTS’ use in the study needs to be refined or better explained.

Response: We have decided to use a different mnemonic that aligns better with scoping reviews. The “PCC” mnemonic is recommended as a guide to construct a clear and meaningful title for a scoping review. The PCC mnemonic stands for the Population, Concept, and Context. There is no need for explicit outcomes, interventions or phenomena of interest to be stated for a scoping review; however, elements of each of these may be implicit in the concept under examination. Please see below for the updated PCC framework, as well as on page 6, 2nd paragraph. 

Eligibility: “For all searches, studies were included or excluded based on the Population, Concept, and Context (PCC) framework for scoping reviews [31]. As such, the participant population was defined as food insecure populations with diabetes (prediabetes, type 1 or 2, or gestational); the concept was recommendations, practices, strategies, or interventions of any nature that addressed diabetes self-management in a food insecure population; studies of all contexts were considered with no specifications for timing and setting. Studies of all designs were acceptable. The studies needed to be published in English for review.”

2. The search conducted until the end of November 2018, retrieved 3066 articles (Fig 1)

[29] “ – Not clear why the citation is included here.

Response: The citation has been deleted.

3. “Twenty-one articles were selected for review based on the inclusion criteria”. May be better to move this sentence to after, "acceptable articles were reviewed in full..."

Response: The sentences have been reworded and moved as suggested. 

4. “Full text-articles were necessary to be included in this review”. This needs some clarity and could be reworded.

Response: Changed to: “Only full text-articles were included in this review.”

5. In the result section, the statement, “Although research about families of children with diabetes is sparse, some recommendations for adults with diabetes can be adopted for children, appears to be more conclusion and discussion than result. Might be better to move it to discussion or conclusion.

Response: We have deleted this sentence based on a previous reviewer’s comments. 

6. I am not sure why the discussion section was presented in sub-titles. It may make for better reading if the subtitles are removed and the discussion given a good flow with a paragraph discussing each point stated in the present subtitles.

Response: We have kept the subtitles in as we think it is easier for our readers (primarily care providers) to follow or direct their attention to the sections that they are most interested in.

7. The conclusion seems lengthy, making it difficult to follow key take away from the study which was left till the last sentence. It might improve the article if more concise conclusion relevant to the objective of the article is written at the beginning and other less relevant sentences written later of left out entirely. The last sentence has the most essential conclusion (Also refers to the study aim) and should be the focus of the whole section.

Response: We have taken the reviewers comments and revised our conclusion. Please see our revisions below and on page 36.

Conclusion: Clinicians can adopt several strategies to better support diabetes self-management among food insecure populations. Routine household food insecurity screening is a logical first step, followed by tailoring of diabetes management plans and interventions via medication management, community referrals, assessing coping strategies, supportive care provider-patient relationships, and smoking cessation. However, given the lack of studies, especially outside North America and in populations with gestational and prediabetes, more studies that evaluate the effectiveness of the identified emerging practices are needed to better inform health care providers and provide a global perspective.

---

## [Editor Report · Decision Letter 1]

18 Sep 2019

PONE-D-19-15760R1

Emerging practices supporting diabetes self-management among food-insecure adults and families: A scoping review

PLOS ONE

Dear Dr. Gucciardi,

Thank you for submitting your manuscript to PLOS ONE. After careful consideration, we feel that it has merit but does not fully meet PLOS ONE’s publication criteria as it currently stands. Therefore, we invite you to submit a revised version of the manuscript that addresses the points raised during the review process.

The ammendment required is a better statement in the abstract and conclusion regarding the limitations of the methods used that limited the ability to identify papers from low-and-middle income countries as a consequence of the language of publication, databases used and restricted to diabetes only.

We would appreciate receiving your revised manuscript by Nov 02 2019 11:59PM. To enhance the reproducibility of your results, we recommend that if applicable you deposit your laboratory protocols in protocols.io, where a protocol can be assigned its own identifier (DOI) such that it can be cited independently in the future. For instructions see: http://journals.plos.org/plosone/s/submission-guidelines#loc-laboratory-protocols

We look forward to receiving your revised manuscript.

Kind regards,

David Alejandro González-Chica, Ph.D., M.D.

Academic Editor

PLOS ONE

Additional Editor Comments (if provided):

Dear Enza Gucciardi and co-authors,

I appreciate the efforts made to improve the paper and I believe the reviewers' suggestions were a great contribution. However, I understand and agree with the comments presented by them regarding the limitation to studies in the United States and Canada. Inlcuing this issue in the limitation seems appropriate, but my concern is that such statement is just a little comment at the end of the paper that will be imperceptible for most readers. I have the conviction this paper will have a global impact and maybe missinterpreted by readers from low-and-middle income countries, and your manuscript does not include evidence from these countries. There are two decisions made by the authors that, in my perspective, are the main drivers of this limitation:

1. Studies written in English only: at least in South America, there is a vast literature on this topic that does not have the aim to achieve international journals and are written in Portuguesse or Spaninsh (studies from Brazil, Mexico, Chile representing a total of 350 million inhabitants - as big as Canada and United States together) and published in journals from these countries as their objective is to provide evidence at a national level that may influence health and nutritional policies. Once the large amount of morbidity and deaths due to NCDs such as diabetes AND food insecurity is concentrated in low-and-middle income countries (it can be as high as 87% - see http://www.scielo.br/pdf/csc/v22n2/1413-8123-csc-22-02-0637.pdf), the decision of focusing on papers published in English only is a huge limitation of the study that should be worth mentioning in the abstract and conclusion as well. Therefore, although further research is nedded as stated in both sections of the manuscript, you can not affirm there is "lack of studies, especially outside

North America". These studies exist, but your study was unable to identify them because of the search strategy. For example, if you include the latinoamerican base LILACS (http://lilacs.bvsalud.org/en/) and use the terms "autocuidado diabetes" you would find 450 papers in the topic of diabetes self-management. As I mentioned before, most low-and-middle income countries live in food insecurity. Thus, using that term would exclude a good proportion of papers that would fit your paper aims.

2. As stated in one of your answers to the reviewers "The second paper mentioned above was used in our discussion but was not eligible in our review as it included people with

various chronic diseases, in addition to diabetes. We limited our review to those with diabetes." International studies have shwon that around 30-60% of adults are affected by two or more chronic conditions (prevalence increases with age). Therefore, excluding studies evaluating sample of patients affected by diabetes only would be making your findings less generalizable and probably reflecting strategies targeting younger age grous. The table with the main findings does not include details on the age of the participants to evaluate this.

I am not suggesting that your group reasseses the methods or re-write the paper entirely. I am just requesting a better statement in the abstract and discussion section that the lack of studies outside North America does not mean there are not studies in other countries, but was a limitation imposed by the research strategy (i.e. English only and restricted to diabetes but no other conditions).

Otherwise, this paper is an excellent contribution to the scientific literature.

Kind regards

---

## [Author Response · Author response to Decision Letter 1]

24 Sep 2019

Response: We would like to thank the editor for their thoughtful and constructive feedback and the opportunity to strengthen the manuscript and resubmit our manuscript. Please find below our responses to the editor’s comments. Thank you.

Dear Enza Gucciardi and co-authors,

I appreciate the efforts made to improve the paper and I believe the reviewers' suggestions were a great contribution. However, I understand and agree with the comments presented by them regarding the limitation to studies in the United States and Canada. Inlcuing this issue in the limitation seems appropriate, but my concern is that such statement is just a little comment at the end of the paper that will be imperceptible for most readers. I have the conviction this paper will have a global impact and maybe missinterpreted by readers from low-and-middle income countries, and your manuscript does not include evidence from these countries. There are two decisions made by the authors that, in my perspective, are the main drivers of this limitation:

1. Studies written in English only: at least in South America, there is a vast literature on this topic that does not have the aim to achieve international journals and are written in Portuguesse or Spaninsh (studies from Brazil, Mexico, Chile representing a total of 350 million inhabitants - as big as Canada and United States together) and published in journals from these countries as their objective is to provide evidence at a national level that may influence health and nutritional policies. Once the large amount of morbidity and deaths due to NCDs such as diabetes AND food insecurity is concentrated in low-and-middle income countries (it can be as high as 87% - see http://www.scielo.br/pdf/csc/v22n2/1413-8123-csc-22-02-0637.pdf), the decision of focusing on papers published in English only is a huge limitation of the study that should be worth mentioning in the abstract and conclusion as well. Therefore, although further research is nedded as stated in both sections of the manuscript, you can not affirm there is "lack of studies, especially outside

North America". These studies exist, but your study was unable to identify them because of the search strategy. For example, if you include the latinoamerican base LILACS (http://lilacs.bvsalud.org/en/) and use the terms "autocuidado diabetes" you would find 450 papers in the topic of diabetes self-management. As I mentioned before, most low-and-middle income countries live in food insecurity. Thus, using that term would exclude a good proportion of papers that would fit your paper aims.

Response: Thank you for your thoughtful comment and we agree to add more context given the limitation of our search strategy for not including non-English studies. This was added in our abstract and limitation section as suggested. Please see the track changes in the manuscript and text below:

Abstract

“A major limitation of this review is the lack of global representation considering no studies outside of North America satisfied our inclusion criteria, due in part to the English language restriction.”

Limitation section

“The lack of identified studies outside North America can be partly attributed to the English language inclusion criteria”.

2. As stated in one of your answers to the reviewers "The second paper mentioned above was used in our discussion but was not eligible in our review as it included people with

various chronic diseases, in addition to diabetes. We limited our review to those with diabetes." International studies have shwon that around 30-60% of adults are affected by two or more chronic conditions (prevalence increases with age). Therefore, excluding studies evaluating sample of patients affected by diabetes only would be making your findings less generalizable and probably reflecting strategies targeting younger age grous. The table with the main findings does not include details on the age of the participants to evaluate this.

I am not suggesting that your group reasseses the methods or re-write the paper entirely. I am just requesting a better statement in the abstract and discussion section that the lack of studies outside North America does not mean there are not studies in other countries, but was a limitation imposed by the research strategy (i.e. English only and restricted to diabetes but no other conditions).

Otherwise, this paper is an excellent contribution to the scientific literature.

Response: We hope we have addressed your concern with the added statements in the abstract and limitation section regarding studies outside of North America. We understand the major limitation of our paper.

Our goal was to be more specific to inform diabetes practice as diabetes is a complex set of self-management protocols including food (beyond healthy eating, but consistent carbohydrate load throughout the day) medications to take or not take, to name a few, which may not be applicable to other chronic conditions. Given practice guidelines provide information based on disease state we wanted to supplement practice guidelines with emerging recommendations for those challenged by food insecurity.

---

## [Editor Report · Decision Letter 2]

3 Oct 2019

Emerging practices supporting diabetes self-management among food insecure adults and families: A scoping review

PONE-D-19-15760R2

Dear Dr. Gucciardi,

We are pleased to inform you that your manuscript has been judged scientifically suitable for publication and will be formally accepted for publication once it complies with all outstanding technical requirements.

With kind regards,

David Alejandro González-Chica, Ph.D., M.D.

Academic Editor

PLOS ONE
---

## [Editor Report · Acceptance letter]

10 Oct 2019

PONE-D-19-15760R2 

Emerging practices supporting diabetes self-management among food insecure adults and families: A scoping review 

Dear Dr. Gucciardi:

I am pleased to inform you that your manuscript has been deemed suitable for publication in PLOS ONE. Congratulations! Your manuscript is now with our production department. 

With kind regards,

on behalf of

Dr. David Alejandro González-Chica 

Academic Editor

PLOS ONE